# STRIDE: Structure-guided Generation for Inverse Design of Molecules

## Abstract

Machine learning and especially deep learning has had an increasing impact on molecule and materials design. In particular, given the growing access to an abundance of high-quality small molecule data for generative modeling for drug design, results for drug discovery have been promising. However, for many important classes of materials such as catalysts, antioxidants, and metal-organic frameworks, such large datasets are not available. Such families of molecules with limited samples and structural similarities are especially prevalent for industrial applications. As is well-known, retraining and even fine-tuning are challenging on such small datasets. Novel, practically applicable molecules are most often derivatives of well-known molecules, suggesting approaches to addressing data scarcity. To address this problem, we introduce **STRIDE**, a generative molecule workflow that generates novel molecules with an unconditional generative model guided by known molecules without any retraining. We generate molecules outside of the training data from a highly specialized set of antioxidant molecules. Our generated molecules have on average 21.7% lower synthetic accessibility scores and also reduce ionization potential by 5.9% of generated molecules via guiding.

## 1 Introduction

Machine learning and especially deep learning algorithms for high-throughput generation and characterization of materials are leading to a sea change in computational chemistry and materials science. Data-driven predictive and generative models can lead to automated inverse design or digital lab (see Abolhasani & Kumacheva (2023)), the holy grail of computational materials science. However, due to the broad range of materials with varying amounts of data availability, the effectiveness of deep learning techniques varies greatly among different materials classes. Therefore, translating materials data to numerical forms, referred to hereafter as the representation, is key since then such disparate classes of molecules and materials can share representations and allow for transferable models. Generative Deep Learning is an integral step in enabling inverse design for molecules, and thus representations are integral to molecular design. The key topic to be addressed is the representation of molecules.

Large language models can generate responses to queries in various domains as the prompt is used to condition the output of the model. This is commonly known as in-context learning. Similarly, vision-generative models can be guided with images or text to generate images without the need for re-training or even fine-tuning. This is significantly important as queries and prompts from highly specialized domains are often quite restrictive and fine-tuning, let alone retraining, is not feasible without detriment to the model. As discussed before, such situations are commonplace in material science where classes of useful and desirable molecules are highly specialized and have only a few known instances.

Motivated by the success of foundation models in language and computer vision, we investigate guided molecule generation with in-context guidance on pre-trained models. We identify graph-based generative models as suitable candidates for pre-training and in-context learning pipelines. We propose **STRIDE**, **str**ucture guided generative workflow for **in**verse **de**sign of molecules. We leverage a pre-trained 3D diffusion model, molecular encoding, and substructure-guidance to build an automated, in-context learning-based framework for generating novel molecules in 3D.

The contributions of this work are, we:

- Design a 3D molecule-based diffusion and high-throughput screening-based inverse design workflow to automatically generate molecules and optimize for molecular properties.
- Develop sampling algorithms to guide a pre-trained 3D diffusion model to generate molecules of interest without the need for retraining or a conditional model.
- Develop a novel substructure-guided sampling approach on top of pre-trained diffusion models to generate novel molecules, allowing chemists and materials scientists to sample molecules with desired fragments.
- Demonstrate that these 3D diffusion models generate specialized unseen classes of molecules without retraining or even fine-tuning and our techniques can drastically improve molecular properties such as synthetic accessibility.

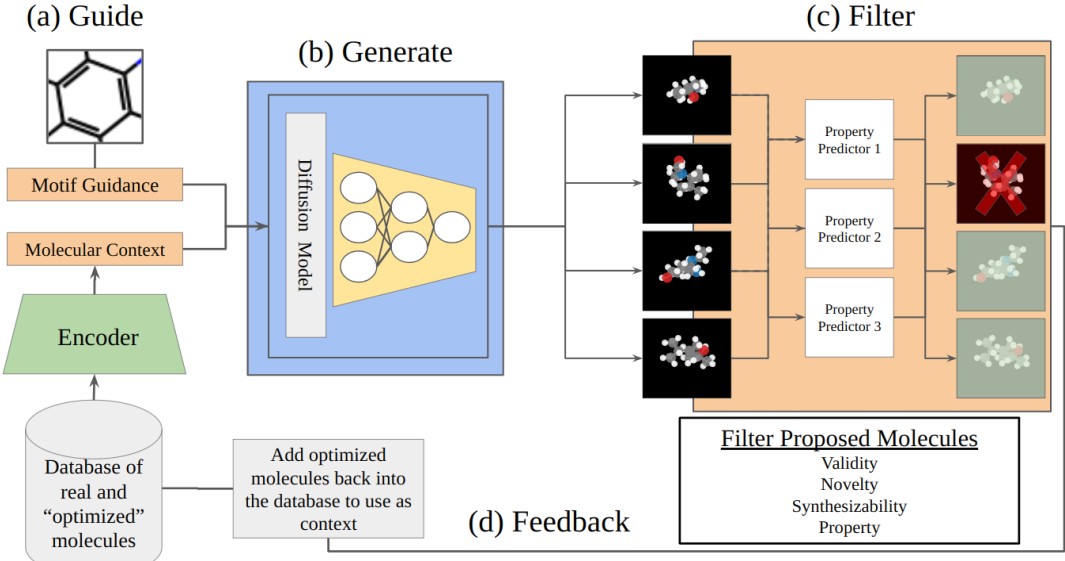

Figure 1: (a) Shows our contribution of prompt selection from a database of known molecules with only a few known samples. A molecular context graph and a user-provided subgraph are inputs to the pre-trained generative model. (b) The pre-trained **generator**, an E(n) Equivariant Diffusion Model, in our case generates new molecules conditioned on the given input. (c) The **filter** stage checks the validity and predicts the properties of generated molecules. (d) Filtered molecules can be added to the database to further guide the generator, creating a **feedback** loop.

We discuss the background representation and model considerations for molecular design workflows in **Section 2**. In **Section 3**, we introduce our molecular generation pipeline **STRIDE**. **Section 4** we demonstrate the application of the workflow on a representative small dataset. **Section 5** and **Section 6** contextualize our contribution and discuss possible shortcomings and future directions, respectively.

## 2 BACKGROUND

The key considerations for designing a molecular generation pipeline are the representations of the molecules, the subsequent choice of deep learning architectures and models, and the inductive biases prevalent in the pipeline as a result. In the following section, we give an overview of the 3D geometric representation, the subsequent Euclidean equivariant model, and the diffusion architecture used in this work.

### 2.1 MOLECULAR REPRESENTATION

The machine-digestible representation of molecules can take many forms. Choosing the correct representation is a key challenge for materials and data scientists, as the most suitable representation

varies depending on the task at hand. In the era of deep learning, the predominant atom-level description of molecules has centered around two graph-based approaches: a strictly topological view of atoms and their connectivity via bonds and a geometric view in 3D space, as a set of vertices with distance weights along the edges. Though both views of a molecule are in some sense interchangeable, as 2D topological representation can be converted to 3D geometric representation using conformer generators. MPNN Gilmer et al. (2017), CGCNN Xie & Grossman (2018), SchNet Schütt et al. (2017), and GemNet Gasteiger et al. (2021), have shown that graph-based networks can achieve state-of-the-art accuracy in molecular property prediction, and molecular dynamics.

in this work, we chose the latter. Similarly, atomic connectivity information from 3D positions can be inferred using bond detection algorithms. Machine learning-based generative models are capable of proposing molecules orders of magnitude faster than what can be screened with expensive computational first principle methods such as DFT. In order to select molecules of interest and filter out proposed molecules, inverse design workflows rely on high-throughput characterization algorithms to select candidate molecules that can be subsequently studied with more computationally expensive methods. Established high-throughput screening algorithms often rely on both geometric and topological representations of molecules so interchangeable representations are desirable as there are well-established methods for molecular property prediction.

## 2.2 E(n) Equivariant Graph Neural Network

E(n) equivariant graph neural networks Satorras et al. (2021) (EGNN)s are special graph neural networks that are equivariant to rotations, translations, reflections, and permutations, i.e. the Euclidean group on N-dimensional Euclidean space, E(n), and permutations. EGNNs generally comprise multiple equivariant graph convolution layers (EGCL) which operate on graphs embedded in n-dimensional Euclidean space. They are a natural choice for our 3D geometric representations.

For a graph $\mathcal{G} = (\mathcal{V}, \mathcal{E})$ with vertices $v_i \in \mathcal{V}$ and edges $e_{ij} \in \mathcal{E}$, in additional to the canonical vectors $h_i \in \mathbb{R}^m$ and $a_{ij} \in \mathbb{R}^k$ representing the vertex and edge features, each vertex also contains $x_i \in \mathbb{R}^n$. $x_i$ is an n-dimensional vector representing the coordinate of vertex $v_i$ in an $n$-dimensional space.

Formally, for a translation vector $g \in \mathbb{R}^n$ and rotation or reflection represented by an orthogonal matrix $Q \in \mathbb{R}^{n \times n}$, we have the condition;

$$Q\mathbf{x}^{l+1} + g, \mathbf{h}^{l+1} = EGCL(Q\mathbf{x}^l + g, \mathbf{h}^l) \tag{1}$$

In effect, each graph $\mathcal{G}$ is embedded into this $n$-dimensional space, and each EGCL is equivariant to rotations, reflections, and translations on this space.

Following the notation from Gilmer et al. (2017) and Satorras et al. (2021), an EGCL consists of message, aggregation, and update functions given by:

$$\mathbf{m}_{ij}^l = \phi_e(\mathbf{h}_i^l, \mathbf{h}_j^l, ||\mathbf{x}_i^l - \mathbf{x}_j||^2, \mathbf{a}_{ij}) \tag{2}$$

$$\mathbf{x}_i^{l+1} = \mathbf{x}_i^l + C \sum_{j \in \mathcal{N}_i} (\mathbf{x} - \mathbf{x}) \phi_x(\mathbf{m}_{ij}) \tag{3}$$

$$\mathbf{m}_i = \sum_{j \in \mathcal{N}_i} \mathbf{m}_{ij} \tag{4}$$

$$\mathbf{h}_i^{l+1} + \phi_h(\mathbf{h}_i^l, \mathbf{m}_i) \tag{5}$$

Where, $\phi_e, \phi_x, \phi_h$ in expressions 2,3, and 5 are learnable functions, generally parameterized by neural networks. $\mathcal{N}_i$ represents the set of neighbors for vertex $i$. We adopt the notation from Satorras et al. (2021), and denote $\mathbf{z} = [\mathbf{x}, \mathbf{h}]$, where $[\cdot]$ is the concatenation operation, to denote the complete set of node features of each graph.

## 2.3 DIFFUSION MODELS

Score-based generative models progressively reverse a forward process that maps the data distribution $x$ to the normal distribution $\mathcal{N}(\mathbf{0}, \mathbf{I})$. Diffusion models are a particular type of score-based model where the reverse process is modeled as a Markov denoising process. The Markov process is defined for time t = 0, . . . , T where the **forward** diffusion transition is defined by a multivariate normal distribution, for all $0 \leq s < t \leq T$:

$$q(\mathbf{z}_t|\mathbf{z}_s) = \mathcal{N}(\mathbf{z}_t|\alpha_{t|s}\mathbf{z}_s, \sigma_{t|s}^2\mathbf{I}) \tag{6}$$

where, $\alpha_{t|s} = \frac{\alpha_t}{\alpha_s}$ and $\sigma_{t|s} =$ For the remained of the paper, we use follow variance preserving process (Ho et al. (2020)), where $\alpha_t^2 = 1 - \sigma_t^2$. We can express $\mathbf{z}_t$ as a linear combination of initial vector $\mathbf{z}_0$ and noise variable $\epsilon$ using the reparameterization trick:

$$\mathbf{z}_t = \alpha_t\mathbf{z}_0 + \sigma_t\epsilon \qquad \epsilon \sim \mathcal{N}(\mathbf{0}, \mathbf{I}) \tag{7}$$

For $\alpha_T \to 0$, this describes a noising process of $\mathbf{z}_0$ such that $\mathbf{z}_T$ is identical to Gaussian distribution centered at 0 with unit variance.

The **generative denoising process** inverts the above **forward** process approximating $z_0$ using a neural network $\hat{\mathbf{z}} = \phi_\theta(\mathbf{z}_t, t)$, at each time step, $t$.

The reverse process with a fixed prior $p(z_T) = \mathcal{N}(0, I)$ is given by:

$$p(\mathbf{z}_{t-1}|\mathbf{z}_t, \mathbf{z}_0) = \mathcal{N}(\mu_{t \to s}(\mathbf{z}_t, \mathbf{z}_0)), \sigma_{t \to s}^2\mathbf{I}) \tag{8}$$

Following the eq. 8, one can iteratively reverse diffusion trajectory by sampling $\mathbf{z}_s$ from $\mathbf{z}_t$ via:

$$\mathbf{z}_s = \frac{1}{\alpha_{t|s}}\mathbf{z}_t - \frac{\sigma_{t|s}^2}{\alpha_{t|s}\sigma_t}\phi_\theta(\mathbf{z}_t, t) + \sigma_{t \to s}\epsilon \qquad \epsilon \sim \mathcal{N}(\mathbf{0}, \mathbf{I}) \tag{9}$$

Following Song et al. (2020); Ho et al. (2020), the diffusion model $\phi_\theta$ can be optimized by the simplified objective:

$$L(\phi_\theta) := \sum_{t=1}^{T} \mathbb{E}_{\mathbf{z}_0 \sim q(\mathbf{z}_0), \epsilon_t \sim \mathcal{N}(0, \mathbf{I})} \left[ || \phi_\theta(\mathbf{z}_t, t) - \epsilon_t ||_2^2 \right] \tag{10}$$

## 3 STRIDE

We propose STRIDE, a molecular generation workflow designed for generating exotic molecules using a pre-trained 3D molecular generator. STRIDE consists of a pre-trained generative model and multiple filtering models. The generation process can be run in a continuous loop and is segmented into 3 stages: **Generate**, **Filter**, **Feedback** as shown in Fig. 1. .

### 3.1 PRE-TRAINED DIFFUSION MODEL

The backbone of **STRIDE** is a graph-based generative model. As shown in Fig. 1, STRIDE uses a pre-trained generative model that can be "prompted" or guided to perform low-temperature sampling. Similar to text-to-image systems, one of the key components of **STRIDE**, is a frozen generative model. But unlike text-to-image models, the prompt encoding is not performed by a large language model. Furthermore, it is infeasible to obtain a universal conditioning label for different classes of molecules, so the generator must remain unconditional. For molecular data, the structure is a significant indicator of performance and of interest, so the motif in itself is a graph. As a consequence, the generation process is similar in spirit to image-guided image generation and in-painting models.

While we focus on 3D equivariant graph diffusion models in this work, other graph-based models such as Gebauer et al. (2019) can be used as well. For the remaining sections of the work, we use the symbol, $\phi_\theta$, to denote a pre-trained graph-based generative diffusion model.

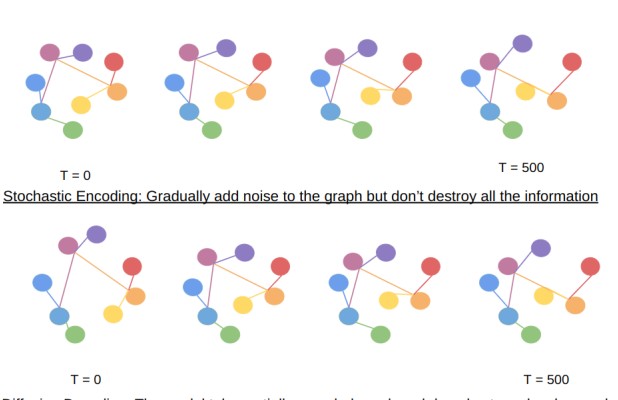

T = 0          T = 500

Stochastic Encoding: Gradually add noise to the graph but don't destroy all the information

T = 0          T = 500

Diffusion Decoding: The model take partially encoded graph and decodes to molecular graph

(a) Partial diffusion guidance only partially "destroys" the original molecule, allowing the denoising network to recover similar molecules

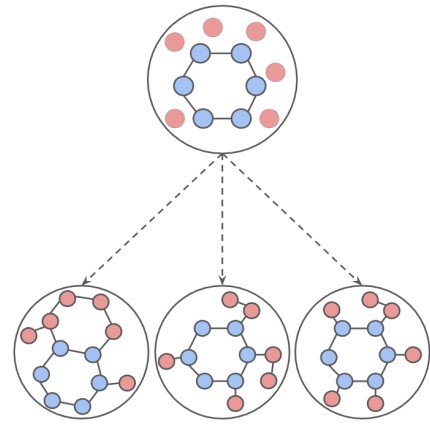

(b) Substructure or Motif conditioning ensures certain parts of the subgraph are guaranteed to exist in the final generated molecule.

Figure 2: Guidance-strategies for pre-trained 3D molecular diffusion models

## 3.2 MOLECULAR CONDITIONING

A major concern for machine-aided molecular generation pipelines is the synthesizability of generated molecules. While de novo molecular generation and optimization techniques have been proposed and shown promising results on multi-objective optimization for specific targets, Gao & Coley (2020) shows that the utility of these approaches is hampered as molecules generated by state-of-the-art generative models cannot be readily synthesized.

Instead of using our pre-trained diffusion model for *de novo* molecular generation, we condition the generator on known functional molecules to leverage external knowledge of synthesizability and industrial processes. The properties of molecules are intrinsically related to it's structure. Therefore unlike images, it is possible to condition molecular generation without additional labels or classifiers. We use known molecular structures to guide pre-trained diffusion models to generate molecules of interest.

Brock et al. (2018) and Kingma & Dhariwal (2018) have shown that "low-temperature" or truncated variance sampling can be used to generate high-fidelity, low-diversity images. Ho & Salimans (2022) show that diffusion models can be guided similarly with a combination of conditional model and unconditional model. As the structure of the generated molecule informs its fidelity, we can similarly guide the molecular generator by truncating the Markov chain process. In the next section, we describe the low-temperature sampling techniques for diffusion models and substructure conditioning to guide the pre-trained generative model.

### 3.2.1 SAMPLING METHODS

We introduce alternative sampling methods from our pre-trained diffusion model which provide varying levels of stochasticity in the generation process. In the generation procedure detailed in Ho et al. (2020), the diffusion process (DDPM) is reversed from a latent encoding at time $t$. Performing the forward process for $t = T$ timesteps leads to all information of $x_0$ being lost. Thus, we perform partial diffusion sampling by sampling $s$ such that $0 \leq s \leq T$ where T is the maximum number of diffusion steps. Using Equation 6

**Implicit Sampling Processes** Song et al. (2020) introduced a reformulation of the forward and generative processes in 6 and 9 which are deterministic. This ensures that the generated molecules $z_0$ depend only on the initial state $z_T$. Unlike the DDPM sampling strategy, this allows for semantic interpolation and conservation of high-level characteristics in the latent state $z_T$.

$$\mathbf{z_s} = \left(\frac{1}{\alpha_{t|s}}\right)(\mathbf{z_t} - \sigma_t \phi_\theta(\mathbf{z}_t, t)) + \sigma_s \phi_\theta(\mathbf{z}_t, t) \tag{11}$$

**Neural ODE Reformulation** Rearraning Eq. 11, we can see that the sampling procedure can be viewed as a differential equation with

$$\frac{\mathbf{z}_t}{\alpha_t} - \frac{\mathbf{z}_s}{\alpha_s} = \left(\frac{\sigma_t}{\alpha_t} - \frac{\sigma_s}{\alpha_s}\right)\phi_\theta(\mathbf{z}_t, t) \tag{12}$$

We can then reparameterize $\lambda_t = \frac{\sigma_t}{\alpha_t}$ and $\bar{\mathbf{z}}_t = \frac{\mathbf{z}_t}{\alpha_t}$ to arrive at the ODE:

$$d\bar{\mathbf{z}}(t) = \phi_\theta\left(\frac{\bar{\mathbf{z}}_t}{1 + \lambda_t^2}\right)d\lambda(t) \tag{13}$$

As mentioned in Song et al. (2020), in the continuous case, equation 12 is the first-order Euler approximation of the ODE in equation 13. For $s \approx t$, we can use the approximation $\phi_\theta(\mathbf{z}_s, s) \approx \phi_\theta(\mathbf{z}_t, t)$, to obtain a non-stochastic iterative encoding process with:

$$\mathbf{z}_t = \alpha_{t|s}\mathbf{z}_s + \left(\sigma_t - \sigma_s\alpha_{t|s}\right)\phi_\theta(\mathbf{z}_s, s) \tag{14}$$

### 3.2.2 MOTIF GUIDANCE

In order for proposed molecules to be practically applicable they must be stable and synthesizable. Ideally, proposed molecules can be synthesized with existing industrial processes based on well-known reaction pathways. Often new practical materials are derivatives of existing, known molecules. To improve our molecular guidance workflow and add further user-controllability, we develop a motif-based guidance, to generate molecules with guaranteed substructures. A conditioning molecule is split into 2 or more substructures. The desired substructures or motifs are masked with $mask_{\text{motif}}$. The non-masked atoms are treated as free atoms and are placed using the generator. The reverse process in Eq. 9 is updated with:

$$\mathbf{z}_{t-1} = mask_{\text{motif}} \odot \mathbf{z}_{t-1} + (1 - mask_{\text{motif}}) \odot \hat{\mathbf{z}}_{t-1} \tag{15}$$

where, $mask_{\text{motif}}$ is 1 if the atom is part of the motif and 0 if it is a free atom. $mask_{\text{motif}} \odot z_{t-1}$ is calculated using the forward process in Eq. 6 or it's variants detailed above. The free atoms are updated using the denoising neural network. It is important to note that the denoising steps on the entire graph rather than just the free atoms. In effect, the generation task is reformulated as a graph completion task with a frozen subgraph. The pseudo-code for the sampling procedure is provided in Appendix C Alg 4.

## 4 RELATED WORK

The first key challenge in generating molecules is of course designing molecules with valid valence structures that obey the laws of chemistry. Topological models are ideal for capturing valence rules as the rules are based on the discrete connectivity of atoms in a molecule. Gómez-Bombarelli et al. (2018) use deep generative models to generate new SMILES (Simplified Molecular Input Line Entry Specification) strings. Kusner et al. (2017) extend such generative algorithms by imposing syntactic and semantic constraints on the decoder such that generated molecules are always topologically valid. While generating valid structures, the SMILES and connectivity-based representation is inherently unable to capture many of the physical invariances present in molecules Guimaraes et al. (2017). Steric effects are difficult to capture as spatial and geometric forces are not evident in these representations. Rotational and permutation symmetries are also not present in such representations. As a result, theoretical molecules proposed with these methods are often unsynthesizable. Furthermore, other classes of materials such as periodic crystals also pose difficulties for SMILES, SELFIES, and other character-based representations as transfer learning is unfeasible.

Graph-based representations of molecules have also been used for molecular generation tasks. Jin et al. (2018) use graph-structured encoder and junction tree-based decoder to generate molecular structures. While graph-structured generative models as introduced by De Cao & Kipf (2018) allow for a greater representation of long-range atomic interactions and provide geometric information

about the molecule. Thus our work is is built on the class of 3D diffusion models introduced by Hoogeboom et al. (2022). Our sampling algorithms follow the literature in image-based diffusion models by Ho et al. (2020); Song et al. (2020); Kingma et al. (2021). The substructure sampling algorithms are inspired by the in-painting and outpainting algorithms using diffusion models by Lugmayr et al.. Concurrently to our work, Runcie & Mey (2023) also proposes a guided diffusion for molecular generation using iterative latent variable refinement by Choi et al. (2021). This is analogous to the substructure guiding case, although the guiding coefficients required to converge are low.

## 5 EXPERIMENTS

### 5.1 EXPERIMENT SET UP

We use an unconditional EDM as the molecular generator for our experiments. The model is trained using the QM9 dataset from Ramakrishnan et al. (2014). The network architecture and the training hyper-parameters are provided in Appendix A. We use a rule-based bond detection algorithm as well as the **DetermineBonds** function provided in RDKIT Landrum (2013). Furthermore, we use two additional pre-trained neural networks, AIMNet-NSE Zubatyuk et al. (2021) and ALFABET St. John et al. (2020) for ionization potential (IP) and bond disassociation energy (BDE) predictions, respectively. AIMNet-NSE significantly requires 3D positions of the molecule for predicting where whereas ALFABET requires atom connectivity and bond order information. We also use RDKit to measure the synthetic accessibility (SA) score Ertl & Schuffenhauer (2009) of the generated molecules .

### 5.1.1 TARGET DATASET

As mentioned above, conditioning on small datasets generated by experts is of particular importance for the practical applicability of deep learning-based generative workflows. We select 29 publicly available antioxidants in the AODB dataset by Deng et al. (2023) from the anti-oxidant dataset as detailed by Moussa et al. (2023). Antioxidants inhibit oxidation and have various industrial applications. Significantly, such a small disparate dataset, as compared with the pre-training dataset, provides an important case study. In many cases, materials engineers are tasked to sample from this sparse subset of the dataset. The entire dataset is reproduced in Appendix B

### 5.2 GUIDED GENERATION RESULTS

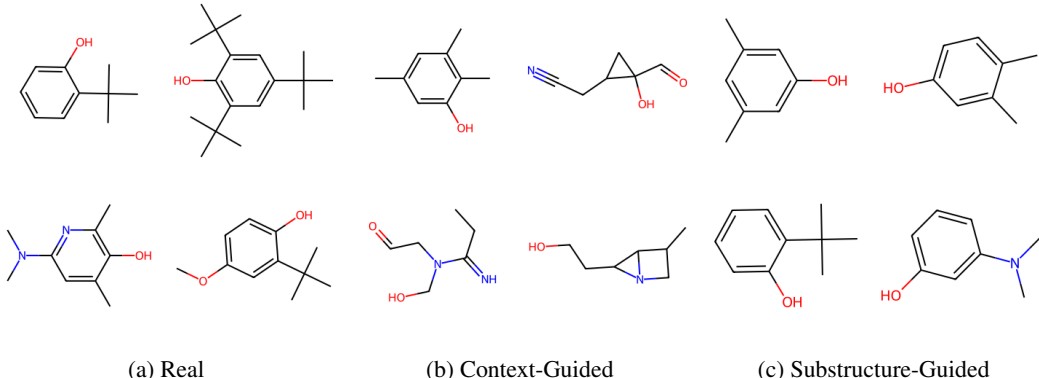

|         (a) Real          |     (b) Context-Guided     |   (c) Substructure-Guided   |

Figure 3: Visualizations of generated molecules with guiding algorithms.

We shift the distribution of generated molecules of the pre-trained model using the sampling methods based on Equations. 9 and 11 and the partials diffusion process. Significantly, in Fig. 4 we show improved SAScores.

The diversity of the context molecules is a concern as so few samples are available. As a result, we also turn on the feedback loop to take highly synthesizable (SA score ¡ 3) and low ionization

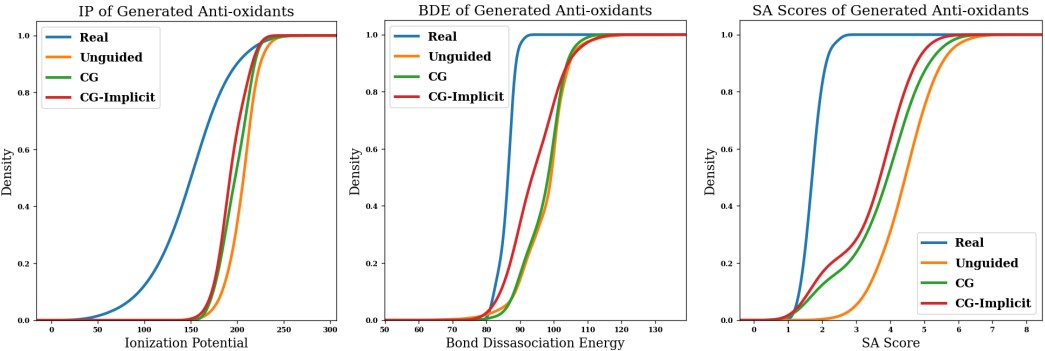

Figure 4: STRIDE is able to significantly shift the distribution of generated molecules by performing partial diffusion sampling on the real dataset. We see improved SAScore and ionization potentials when guiding with the small target dataset.

potential (¡ 165 kcal/mol) molecules and reuse them as feedback to generate molecules. Figure 5 shows a slight improvement in generated molecules properties.

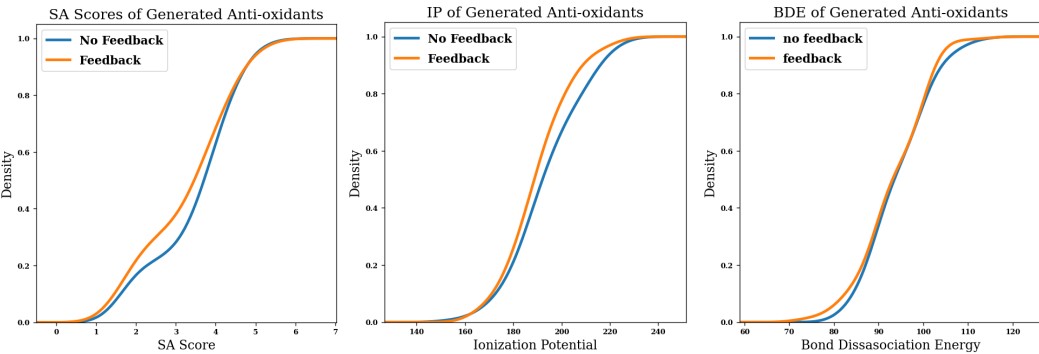

Figure 5: Figure of diffusion dynamics for the different encoding and sampling methods

# 6 DISCUSSION AND FUTURE WORK

A major concern for the generative model used is the full $N^2$ adjacency matrix required for the generation process and the bond inference. This is of particular concern when scaling up to large molecules such as polymers which may have hundreds of atoms per molecule. The adjacency matrix is equivalent to the attention matrix prevalent in modern deep-learning models. Thus the optimization techniques for both full and sparse attention matrices may be applied in these models as well. Furthermore, improvements in diffusion models over discrete probabilities may alleviate issues.

Our current experiments show a significant decrease in model efficiency, as many of the guided molecules are invalid. We believe this is due to the small size of the pre-training dataset as well as the exceptionally few context molecules. We plan on re-training the backbone model for STRIDE using significantly more data. Furthermore, recent work in 3D + 2D graph diffusion models by Vignac et al. (2023), Zhang et al. (2023), and Peng et al. (2023) show significant improvements in generation capabilities and highlight the need for inverse design systems built around such models.

The limitations of the transferability of models and representation across material types and chemistries from distinct domains limit the application of cutting-edge deep learning techniques to materials science. New and exciting materials often require knowledge and experimentation on sparsely distributed and highly self-similar data. Ideally, foundational models for chemistry and materials would be trained on large swaths of available data and would generalize on materials from

unseen domains. Such models require carefully chosen representations derived from natural laws and architectures that are transferable across materials. Akin to the lowest common multiple between a set of numbers, the features used to represent the data must be present entire data domain of the models. Since we show that in-context learning can be enabled on 3D equivariant diffusion models and generate targeted molecules, such models present a promising avenue for a viable foundation model for chemistry.

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

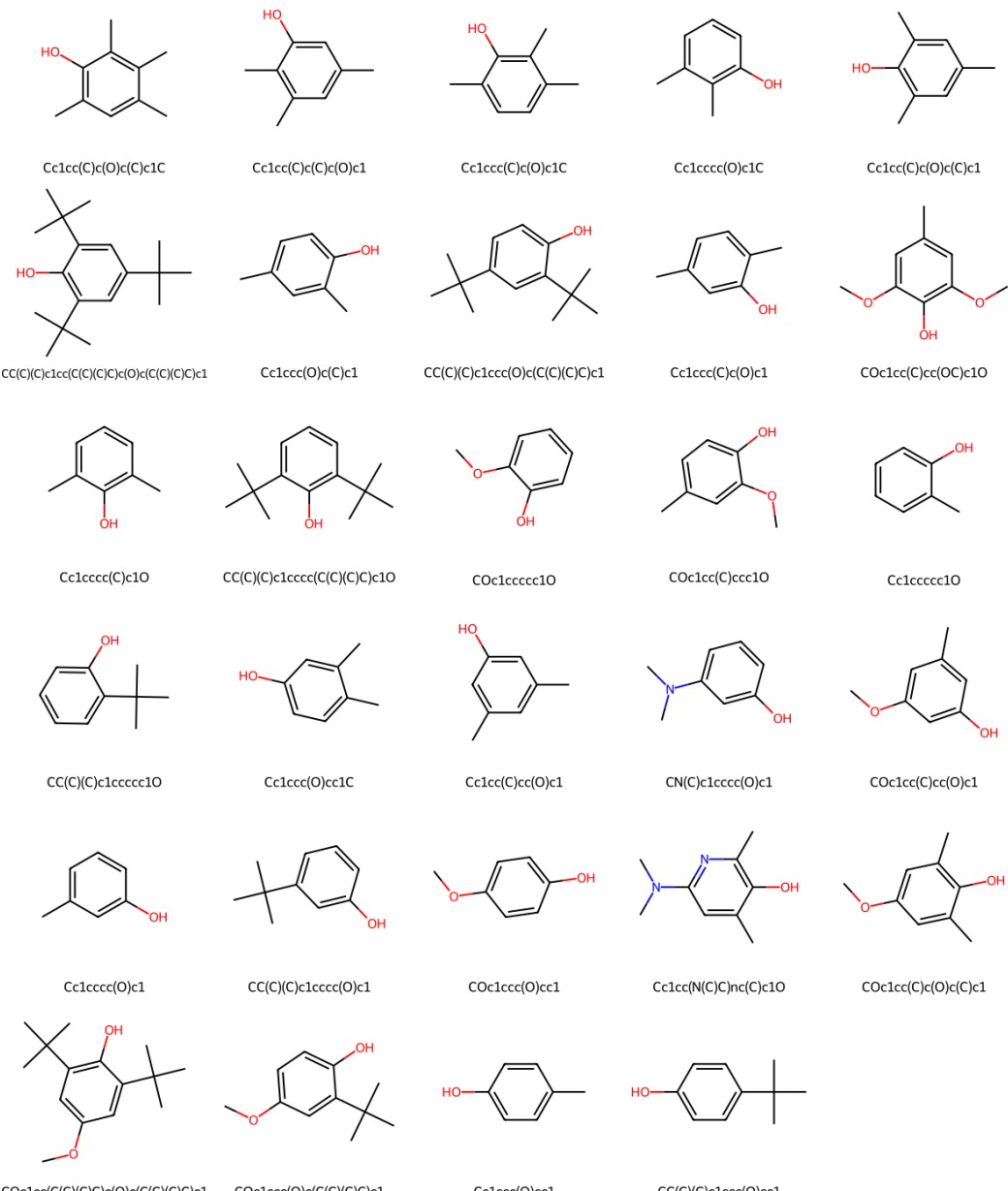

Figure 6: Subset of Anti-oxidant data in AODB used as the target dataset

# A  PRE-TRAINING DETAILS

We use an E(3)-equivariant neural network to approximate the backward process of our diffusion model. We use a 9-layer EGCL, where learnable functions $\phi_e, \phi_x, \phi_u$ in 2 are approximated using by multi-layer perceptions.

## A.1  EDM TRAINING

**EDM Hyperparameters** The EDM network architecture consists of a 9-Layer Equivariant GNN. Further details of each operation can be found in Appendix B in Hoogeboom et al. (2022). For our network, we select nf to be 256.

The input node features consist of a 9-element vector. $x \in \mathbb{R}^3$ is the euclidean coordinates of each node. $h_{\text{atom}} \in \mathbb{R}^5$ is the one-hot encoded representation of the atom type. $h_{\text{charge}} \in \mathbb{R}^1$ is the charge of the element. The input to the model is rescaled as $[x, 0.25h_{\text{atom}}, 0.1h_{\text{charge}}]$

**Training Hyperparameters** The network is trained for 3000 epochs on the QM9 dataset using T = 1000 diffusion steps.

## B  TARGET DATASET INFORMATION

The set of molecules used to prompt the workflow and perform analysis can be viewed in Fig. 6.

The validity of the molecules is screened using the following bond distances in Table 1 to determine the bond order. Then valence conditions for each atom are checked to guarantee

| Bond Atoms | Single | Double | Triple | Aromatic |
|:---:|:---:|:---:|:---:|:---:|
| H - H | 74 | | | |
| H - C | 109 | | | |
| H - N | 101 | | | |
| H - O | 96 | | | |
| C - C | 154 | 134 | 120 | 140 |
| C - N | 147 | 129 | 116 | |
| C - O | 143 | 120 | 113 | |
| N - N | 145 | 125 | 110 | 134 |
| N - O | 140 | 121 | | |
| O - O | 148 | 121 | | |

Table 1: Typical bond distances (pm) used for validity calculations

## C  SAMPLING ALGORITHMS GUIDANCE

The sampling algorithm from Hoogeboom et al. (2022) is reproduced here:

---
**Algorithm 1** Probabilistic Sampling from EDM

---
1: **Input:** Data point x, Trained model $\phi_\theta$
2: Sample $z_T \sim \mathcal{N}(\mathbf{0}, \mathbf{I})$
3: **for** $t$ in $T, T-1, \ldots, 1$ where $s = t - 1$ **do**
4: $\quad \epsilon \sim \mathcal{N}(\mathbf{0}, \mathbf{I})$
5: $\quad$ Subtract COG from $\epsilon^{(x)}$ in $\epsilon = [\epsilon^{(x)}, \epsilon^{(h)}]$
6: $\quad \mathbf{z}_s = \frac{1}{\alpha_{t|s}}\mathbf{z}_t - \frac{\sigma_{t|s}^2}{\alpha_{t|s}\sigma_t}\phi_\theta(\mathbf{z}_t, t) + \sigma_{t \to s}\epsilon$
7: **end for**
8: Sample $\mathbf{x}, \mathbf{h} \sim p(\mathbf{x}, \mathbf{h}|\mathbf{z}_0)$

---

---
**Algorithm 2** Implicit Sampling from EDM

---
1: **Input:** Data point x, Trained model $\phi_\theta$
2: Sample $z_T \sim \mathcal{N}(\mathbf{0}, \mathbf{I})$
3: **for** $t$ in $T, T-1, \ldots, 1$ where $s = t - 1$ **do**
4: $\quad$ Subtract COG from $\epsilon^{(x)}$ in $\epsilon = [\epsilon^{(x)}, \epsilon^{(h)}]$
5: $\quad \mathbf{z}_s = \left(\frac{1}{\alpha_{t|s}}\right)(\mathbf{z_t} - \sigma_t\phi_\theta(\mathbf{z}_t, t)) + \sigma_s\phi_\theta(\mathbf{z}_t, t)$
6: **end for**
7: Sample $\mathbf{x}, \mathbf{h} \sim p(\mathbf{x}, \mathbf{h}|\mathbf{z}_0)$

---

---

**Algorithm 3** Partial Diffusion

---

1: **Input:** Data point x, Trained model $\phi_\theta$
2: Sample $S \sim [100, 200, 500, 700]$
3: $\epsilon \sim \mathcal{N}(\mathbf{0}, \mathbf{I})$
4: $z_T = \alpha_S x + \sigma_S$
5: **for** $t$ in $T, T-1, \dots, 1$ where $s = t - 1$ **do**
6:     Subtract COG from $\epsilon^{(x)}$ in $\epsilon = [\epsilon^{(x)}, \epsilon^{(h)}]$
7:     $\mathbf{z}_s = \text{inversion}(\mathbf{z}_t, \phi_\theta)$
8: **end for**
9: Sample $\mathbf{x}, \mathbf{h} \sim p(\mathbf{x}, \mathbf{h} | \mathbf{z}_0)$

---

Here either Algorithm 1 or 2 can be used for inversion.

---

**Algorithm 4** Motif-Guidance

---

1: **Input:** Data point x, Trained model $\phi_\theta$
2: Sample $S \sim [100, 200, 500, 700, 1000]$, motif $m$, motif mask $mask$
3: $\epsilon \sim \mathcal{N}(\mathbf{0}, \mathbf{I})$
4: $z_T = \alpha_S x + \sigma_S$
5: $m_T = \alpha_S m + \sigma_S \epsilon$
6: **for** $t$ in $T, T-1, \dots, 1$ where $s = t - 1$ **do**
7:     **for** $t$ in $1, 2, \dots,$ num_resamples **do**
8:         Subtract COG from $\epsilon^{(x)}$ in $\epsilon = [\epsilon^{(x)}, \epsilon^{(h)}]$
9:         $\mathbf{z}_s^u = \text{inversion}(\mathbf{z}_t, \phi_\theta)$
10:         $\mathbf{z}_s^k = \alpha_S m + \sigma_S \epsilon \, \text{inversion}(\mathbf{z}_t, \phi_\theta)$
11:         $\mathbf{z}_s = mask \odot \mathbf{z}_s^u + (1 - mask) \odot \mathbf{z}_s^u$
12:     **end for**
13: **end for**
14: Sample $\mathbf{x}, \mathbf{h} \sim p(\mathbf{x}, \mathbf{h} | \mathbf{z}_0)$

---

