# OpenReview forum: "STRIDE: Structure-guided Generation for Inverse Design of Molecules"
_NeurIPS.cc/2023/Workshop/AI4Science — NeurIPS2023-AI4Science Poster_

### Official Review · Reviewer_CMWr · 2023-10-22

**Rating:** 4
**Confidence:** 4

**Review:**

**Summary**

This paper builds on recent work on diffusion models for molecular design and shows the use of guidance to tailor design. The authors introduce a workflow to Generate, Filter, and Feedback, where generated examples can be used to guide the next iteration of generation. Under this framework, substructures can be enforced and the authors show that property distribution shifting can be achieved.

**Main Review**

The paper is easy to follow and is motivated by a very relevant problem in molecular design: achieving tailored design with small and scarce datasets. The molecular representation for the diffusion model also enables better compatibility with other classes of molecules beyond organic molecules. Tailored design for various properties is explicitly shown in the Result section, and demonstrates that guiding (and self-guiding) works. However, there are limitations to the approach when placed in the context of existing works and the problem that is being tackled.

**Comments**

1. There are existing works for substructure-constrained generation such as Lib-INVENT [1], Scaffold Decorator [2], DiffSBDD [3, 4] that should be mentioned.

2. The statement: “As a result, theoretical molecules proposed with these methods [in reference to SMILES-based models] are often unsynthesizable”. This is not inherent to the SMILES representation. In the proposed work, optimising SA scores can also be done with SMILES-based models through goal-directed generation. SMILES models have also been coupled with explicit reaction rules, such as in Lib-INVENT [1].

3. SA score is a measure of molecular complexity and is used as a proxy for synthetic feasibility. While it can be indeed useful to optimize for, it should be acknowledged that it is a heuristic.

4. Using pre-trained networks to predict IP and BDE cuts down on oracle evaluation cost, as the alternative would be semi-empirical or DFT calculations which are computational expensive. However, it is important to acknowledge that out-of-distribution prediction (to the training data of these predictors) is a limitation. Molecular generation relies on the accuracy of prediction, especially since the molecules are used for iterative guidance. This problem can be (to an extent) circumvented when using first-principles simulations (like DFT). By using predictor networks, the generalizability of the generative model is inherently constrained.

5. Diffusion models can have relatively long sampling times. How many molecules needed to be sampled and what is the wall time to shift the distributions?

6. Figures should be labelled with the units of energy. From the text, it seems that kcal/mol is used.

7. The authors mention that the current workflow has “significantly decreased model efficiency” because many generated molecules are invalid. Can the authors provide some statistics around this? Validity, Novelty, etc. since these are also metrics listed in Figure 1. Generating invalid molecules sometimes is ok if generation is computationally inexpensive. However, diffusion models can have long sampling times and it is important to gauge the compute required.

8. The authors design a workflow to tackle low-data regime scenarios where one is interested in fixing a molecular scaffold/motif. However, the generator is pre-trained on QM9 which is still decently large (although perhaps small compared to common “drug-like” libraries like ChEMBL and ZINC). Given that the 29 selected antioxidants contain common scaffolds, were any of these scaffolds present in the QM9 dataset?

9. Existing works can explicitly shift distributions by applying optimization algorithms on top. For example, [5] used the REINVENT [6] model to explicitly optimize for various quantum-mechanical properties, where keeping a substructure fixed is possible.

10. Re-training and fine-tuning is indeed challenging with small datasets and to shift distributions. However there are previous works that aim to address this:

**From a distribution learning perspective:**
[7] performs iterative fine-tuning on generated molecules to shift the distribution. They significantly shift the ionisation potential energy distribution. In the current work, the IP energy is given in kcal/mol - 165 was referenced in the text, which is about 7.15 eV. In [7], iterative fine-tuning shifted the distribution by ~4 eV. This is not to say that the current work should also be able to do this. This is to highlight that existing works have tackled this problem and have shown success.

**From a goal-directed generation perspective:**
Language models such as REINVENT can also shift property distributions as shown in [5]. The scaffolds of the 29 selected antioxidants are also not structurally complex and many would be expected to be present in popular open-source datasets such as ChEMBL. The core scaffold is benzene which is one of the most common scaffolds. Given this commonality, pre-training on a large dataset once is sufficient to perform goal-directed generation, which could then explicitly optimize for target properties such as IP energy, SA score, or both simultaneously.

Overall, the work shows that guidance can be applied to the E(3) Diffusion Model (EDM) [8] for tailored molecular design. However, more results are needed to demonstrate the benefit of this approach compared to existing works. For example, showing this can work on molecular modalities other than organic molecules, such as metal-containing compounds where SMILES-based representations struggle.

**Minor Comments**

*  Equation 3 notation: the distance subtraction should have subscripts indicating a pair of coordinates, where $i$ != $j$. This
   operation ensures equivariance to translation and rotation [9].


**References**

[1] Fialková, Vendy, et al. "LibINVENT: reaction-based generative scaffold decoration for in silico library design." Journal of Chemical Information and Modeling 62.9 (2021): 2046-2063.

[2] Arús-Pous, Josep, et al. "SMILES-based deep generative scaffold decorator for de-novo drug design." Journal of cheminformatics 12.1 (2020): 1-18.

[3] Schneuing, Arne, et al. "Structure-based drug design with equivariant diffusion models." arXiv preprint arXiv:2210.13695 (2022).

[4] Harris, Charles, et al. "Flexible Small-Molecule Design and Optimization with Equivariant Diffusion Models." ICLR 2023-Machine Learning for Drug Discovery workshop. 2023.

[5] Li, Cheng-Han, and Daniel P. Tabor. "Generative organic electronic molecular design informed by quantum chemistry." Chemical Science (2023).

[6] Olivecrona, Marcus, et al. "Molecular de-novo design through deep reinforcement learning." Journal of cheminformatics 9.1 (2017): 1-14.

[7] Westermayr, Julia, et al. "High-throughput property-driven generative design of functional organic molecules." Nature Computational Science 3.2 (2023): 139-148.

[8] Hoogeboom, Emiel, et al. "Equivariant diffusion for molecule generation in 3d." International conference on machine learning. PMLR, 2022.

[9] Satorras, Vıctor Garcia, Emiel Hoogeboom, and Max Welling. "E (n) equivariant graph neural networks." International conference on machine learning. PMLR, 2021.

---

### Official Review · Reviewer_ZbE3 · 2023-10-25
**More motivation and evaluation is needed**

**Rating:** 4
**Confidence:** 4

**Review:**

Summary: This paper presents a method called STRIDE for performing molecule design. STRIDE uses a diffusion model to generate 3D structures of molecules and uses a pre-trained model guided by substructures to generate new molecules.

Strengths: The paper tackles the problem of generating molecules in a low-data regime, which is an important problem to solve. The paper presents a novel diffusion model for molecular generation in this regime.

Weaknesses: The results are weak and the evaluation is limited. The proposed methods do not improve the ionization energy or bond dissociation energy compared to the unguided model, and the generated molecules underperform the real molecules (although I may be misinterpreting the results since the authors do not state whether these values should be high or low). The diversity and novelty of the generated molecules are not evaluated. There are also no comparisons to existing methods for molecule generation. The paper does not report important values such as the number of molecules that were generated. It is also unclear how the 3D coordinates for the training molecules were generated, given that molecules have a whole distribution of 3D conformations as opposed to a single 3D shape.

Additionally, the authors partially motivate their work by arguing that substructure guiding improves the synthesizability of molecules. While the SAscores of the guided version of the model are slightly improved, this does not guarantee synthesizability. Using a substructure that is synthesizable and then modifying it can result in a molecule that is very difficult to synthesize.

Furthermore, the particular problem of antioxidant generation does not seem to require the complexity of generative diffusion models. The molecules are very small, and particularly when a substructure is selected and fixed, only a handful of atoms need to be added. At this point, it would likely be possible to simply enumerate and score potential molecules rather than using a diffusion process, and such enumeration and scoring could be faster and could result in a better set of generated molecules. The authors should consider enumeration + scoring as a potential baseline, or they should justify why generative models are needed for such small molecules. Alternatively, the authors could apply their method to designing larger molecules, where a generative model would provide more benefit.

---

### Meta-Review · Area_Chair_gZkj · 2023-10-27

**Recommendation:** Accept (Poster)
**Confidence:** 4

**Metareview:**

&nbsp;

I recommend acceptance of the paper. The reviewers have done an excellent and thorough analysis of the paper, highlighting a number of area in which the empirical evaluation, presentation, and references to the literature, could be improved. As such, I would encourage the authors to try to address all points raised ahead of the workshop, namely:

&nbsp;

1. To provide details on statistics of the diversity and novelty of the generated molecules.
2. To indicate the number of molecules that were generated.
3. To give the wall-clock time for the models used as well as a discussion of the practical implications of the runtime.
4. To describe how 3D coordinates for the training molecules were generated.
5. To consider using enumeration and scoring as a baseline

&nbsp;

Further to the authors’ comment on the synthesizability of generative models using the SMILES representation, the problem with invalid molecules stems not from the representational capacity of SMILES but rather on the interplay between the generative model and the optimization scheme [1]. Indeed, there are many recent examples of string-based approaches such as SMILES and SELFIES being performant on toy tasks [2, 3, 4].

&nbsp;

I look forward to the discussion at the workshop!

&nbsp;

__**References**__

&nbsp;

[1] Griffiths, R.R. and Hernández-Lobato, J.M., 2020. Constrained Bayesian optimization for automatic chemical design using variational autoencoders. Chemical science, 11(2), pp.577-586.

[2] Alperstein, Z., Cherkasov, A. and Rolfe, J.T., 2019. All SMILES variational autoencoder. arXiv preprint arXiv:1905.13343.

[3] Maus, N., Jones, H., Moore, J., Kusner, M.J., Bradshaw, J. and Gardner, J., 2022. Local latent space bayesian optimization over structured inputs. Advances in Neural Information Processing Systems, 35, pp.34505-34518.

[4] Maus, N., Wu, K., Eriksson, D. and Gardner, J., 2023, April. Discovering Many Diverse Solutions with Bayesian Optimization. In International Conference on Artificial Intelligence and Statistics (pp. 1779-1798). PMLR.

&nbsp;